# IVIM-DWI-Based Radiomics for Lesion Phenotyping and Clinical Status Prediction in Relapsing–Remitting Multiple Sclerosis

**DOI:** 10.3390/jcm14196753

**Published:** 2025-09-24

**Authors:** Othman I. Alomair, Mohammed S. Alshuhri, Haitham F. Al-Mubarak, Sami A. Alghamdi, Abdullah H. Abujamea, Salman Aljarallah, Nuha M. Alkhawajah, Yazeed I. Alashban, Nyoman D. Kurniawan

**Affiliations:** 1Radiological Sciences Department, College of Applied Medical Sciences, King Saud University, P.O. Box 145111, Riyadh 4545, Saudi Arabia; salghamdi1@ksu.edu.sa (S.A.A.); yalashban@ksu.edu.sa (Y.I.A.); 2King Salman Centre for Disability Research, Riyadh 11614, Saudi Arabia; 3Radiology and Medical Imaging Department, College of Applied Medical Sciences, Prince Sattam Bin Abdulaziz University, AlKharj 11942, Saudi Arabia; m.alshuhri@psau.edu.sa; 4Glasgow Experimental MRI Centre, Institute of Neuroscience and Psychology, University of Glasgow, Glasgow G61 1QH, UK; haitham_f99@outlook.com; 5Department of Radiology and Medical Imaging, King Saud University Medical City & College of Medicine, King Saud University, Riyadh 7805, Saudi Arabia; abujamea@ksu.edu.sa; 6Department of Medicine, College of Medicine, King Saud University, P.O. Box 145111, Riyadh 4545, Saudi Arabia; saljarallah@ksu.edu.sa (S.A.); nalkhawajah@ksu.edu.sa (N.M.A.); 7Australian Institute for Bioengineering and Nanotechnology, Centre for Advanced Imaging, The University of Queensland, Brisbane QLD 4072, Australia; nyoman.kurniawan@cai.uq.edu.au

**Keywords:** magnetic resonance imaging (MRI), machine learning (ML), relapsing–remitting multiple sclerosis (RR-MS), expanded disability status scale (EDSS), radiomics, intravoxel incoherent motion diffusion-weighted imaging (IVIM-DWI)

## Abstract

**Background/Objectives:** Multiple sclerosis (MS) is an autoimmune disorder affecting the central nervous system, characterised by the degradation of myelin, which results in various neurological symptoms. This study aims to utilise radiomics features to evaluate the predictive value of IVIM diffusion parameters, namely, the true diffusion coefficient (*D*), pseudo-diffusion coefficient (*D**), and perfusion fraction (*f*), in relation to disability severity, assessed using the Expanded Disability Status Scale (EDSS), and mobility in patients with relapsing–remitting MS. **Methods:** This retrospective cross-sectional study analysed MRI data from 197 patients diagnosed with multiple sclerosis (MS). Quantitative intravoxel incoherent motion (IVIM) parameters were obtained using a 1.5 Tesla MRI scanner. Clinical information collected included age, disease duration, number of relapses, status of disease-modifying therapy (DMT), and the need for mobility assistance. Machine learning (ML) techniques, such as XGB, Random Forest, and ANN, were employed to explore the relationships between radiomic IVIM parameters and these clinical variables. **Results:** IVIM radiomics achieved high accuracy in lesion phenotyping. Random Forest distinguished enhancements from non-enhancing lesions with 96% accuracy and AUC = 0.99 with IVIM-*f* and *D** maps. CNN also reached ~92% accuracy (AUC 0.97) with IVIM-*f*. For disability prediction, IVIM-*D* and *D** radiomics strongly correlated with EDSS: Random Forest achieved 89% accuracy (AUC = 0.90), while CNN achieved 90% accuracy (AUC = 0.95). Mobility impairment was predicted with the highest performance—RNN achieved 96% accuracy (AUC = 0.99) across IVIM-*f* features. In contrast, relapse history, disease duration, and treatment status were poorly predicted (<75% accuracy). **Conclusions:** ML analyses of IVIM metrics provided independent predictors of functional impairment and disability in MS. Our novel approach may be used to improve diagnostic accuracy and develop personalised treatment strategies for MS patients.

## 1. Introduction

Multiple sclerosis (MS) is a complex chronic autoimmune disease that attacks the central nervous system, causing inflammation, demyelination, and neurodegeneration [1,2]. This disease is characterised by the development of multifocal lesions within the brain and spinal cord. Over time, many patients accumulate progressive disability, which is commonly assessed using the Expanded Disability Status Scale (EDSS) [3,4,5,6].

A persistent challenge in MS care is the mismatch between MRI findings and clinical presentation [7]. Conventional MRI is considered the gold standard for diagnosing and monitoring MS, but only limited correlations have been found between lesion loads and functional impairment [8,9], with an inability to fully capture the clinical impact of the disease [10]. Sequences such as FLAIR and post-contrast T_1_-weighted imaging have been used as the primary tools for detecting lesions and assessing disease activity, while T_2_-weighted imaging provides additional support in lesion detection [11,12].

However, these sequences are primarily qualitative and offer limited information about underlying tissue damage [7,13]. Gadolinium enhancement is helpful in identifying active inflammatory lesions, but its repeated use raises concerns regarding safety, including gadolinium accumulation in the brain and potential nephrotoxicity [14,15].

Advanced MRI techniques have emerged to address these limitations by providing more detailed insight into tissue microstructure. One such technique is intravoxel incoherent motion (IVIM) imaging, which uses multi-b-value diffusion-weighted imaging to separate true tissue diffusion from perfusion-related pseudo-diffusion. IVIM produces parametric maps that reflect tissue diffusivity (*D*), pseudo-diffusion (*D**), and perfusion fraction (*f*), offering quantitative information about both cellular and microvascular changes [16,17,18]. Lower b-values predominantly reflect capillary perfusion effects, whereas higher b-values are more indicative of true molecular diffusion [19,20].

Previous research conducted by Alomair et al. [17] has demonstrated that these IVIM parameters vary across different types of MS lesions. For example, studies have shown that enhancing, chronic non-enhancing and black hole lesions exhibit distinct IVIM-derived metrics [17]. Black holes, often associated with more severe axonal damage, tend to show markedly elevated diffusion and perfusion values. These observations suggest that IVIM imaging can capture lesion heterogeneity that may not be apparent on conventional scans [17].

Although IVIM maps provide valuable physiological information, their interpretation remains complex, as shown in our previous works [8,17]. Radiomics, the high-throughput extraction of quantitative features from medical images, offers a way to systematically analyse these maps [21,22]. Radiomic features include descriptors of intensity, shape, and texture, which can reveal patterns that are not visually evident. In the context of MS, lesion variability in size, contour, and internal architecture makes radiomics particularly relevant [23,24].

Recent studies have applied radiomics to MS imaging [25,26], where features extracted from conventional MRI sequences were analysed using machine learning techniques to distinguish active from inactive lesions without the need for gadolinium, with the accuracy reaching ~98% [25,26]. Another study by [27] reported classification accuracy for active/inactive lesions with an AUC approaching 0.95 using T_2_-weighted images alone [27]. Preliminary efforts to predict long-term disability from baseline MRI radiomic features have shown moderate success (accuracy ~77% for predicting 10-year high EDSS) [28]. Although the results have been encouraging, model performance has remained moderate, indicating room for improvement.

The aim of this study is to evaluate whether radiomic features derived from IVIM parametric maps can (1) accurately classify lesions as active (gadolinium-enhancing) or non-active, providing an imaging biomarker of lesion activity, and (2) correlate with patients’ clinical metrics, enabling prediction of disability level, mobility impairment, relapse rate, and treatment status.

We hypothesised that integrating radiomics with advanced imaging modalities such as IVIM may yield more powerful tools for characterising MS. IVIM-based radiomics could potentially provide both anatomical and physiological insights, offering a more comprehensive assessment of lesion activity and its clinical implications.

## 2. Materials and Methods

### 2.1. Study Design and Participants

We conducted a retrospective cross-sectional single-time-point observational study on a cohort of relapsing–remitting MS patients enrolled at a single centre. The inclusion criteria were a definite MS diagnosis (2017 McDonald criteria) [29,30], a relapsing–remitting disease course, and the availability of a recent brain MRI including a routine MS imaging protocol and diffusion sequences. The key exclusion criteria were other neurological disorders or MRI artefacts that precluded analysis. A total of 224 patients were recruited, 197 of whom had usable imaging data (27 were excluded due to motion artefacts or not meeting the diagnostic criteria) [8,17]. Figure 1 presents a flowchart of the patient selection criteria and data processing [31,32].

The cohort included both females and males (at an approximately 3:1 ratio, reflecting the MS sex distribution), with a mean age in the mid-30s. The clinical data collected for each patient included the EDSS (Expanded Disability Status Scale) score (mean ~2.3, range 0–8 in our sample), disease duration (years since diagnosis), recent relapse history (number of relapses in the past year), self-reported mobility impairment (e.g., need for assistance or abnormal gait, if any), and current disease-modifying therapy (DMT) status. Table 1 presents the demographic and clinical data of the RR-MS patients. Consent forms were waived due to the retrospective nature of this study. The study protocol was approved by King Saud University Medical City’s Institutional Review Board (IRB No. E-23-7517).

### 2.2. MRI

Brain MRI scans were performed on a 1.5 Tesla GE MRI scanner (GE Healthcare, Waukesha, WI, USA) with an 8-channel phased-array head coil. The sequences included axial T_2_-weighted and FLAIR imaging for lesion detection, 3D T_1_-weighted pre- and post-contrast (0.1 mmol/kg gadolinium) imaging for identifying enhancing lesions, and diffusion-weighted imaging (DWI) with multiple b-values for IVIM analysis [17]. All the post-contrast T1-weighted images were acquired within 10 min after administering the contrast agent.

The DWI sequence was single-shot echo-planar imaging with b-values of 0, 30, 50, 70, 100, 200, 500, and 1000 s/mm^2^ applied in 3 orthogonal directions [33]. From the DWI data, IVIM model fitting was performed voxel-wise using a bi-exponential decay model to generate parametric maps of ADC, *D*, *D**, and *f* as previously reported by Alomair et al. [17]. Low b-values (<200 s/mm^2^) were included to capture perfusion-related contributions, whereas higher b-values were required to quantify true molecular diffusion and ADC [8,17].

### 2.3. Lesion Segmentation and Classification

All multiple sclerosis (MS) lesions were identified, assessed, and classified using conventional MRI sequences, primarily axial T_2_-weighted and FLAIR images, and confirmed by two board-certified neuroradiologists, each with more than 10 years of clinical experience in neuroimaging. The number of lesions was recorded for each patient as part of the lesion burden analysis. Regions of interest (ROIs) were manually delineated on post-contrast axial T_1_-weighted images using ITK-SNAP software (version 4.2.2) [34], with the aid of co-registered T_2_-weighted images to ensure precise localisation and anatomical consistency [35,36,37]. ROIs were carefully drawn to capture representative lesions while avoiding areas affected by partial volume effects, motion artefacts, or overlapping anatomical structures. To ensure uniformity across the dataset, all ROI annotations were performed by the same neuroradiologist, who conducted the lesion analysis for all patients. Lesions were classified into three categories—gadolinium-enhancing (active), non-enhancing (chronic), and T_1_-hypointense "black hole" lesions—reflecting different stages of pathological evolution. Additional details regarding the imaging protocol and lesion segmentation strategy are available in our previously published study [17].

### 2.4. Extraction and Selection of Radiomic Features

Figure 2 illustrates the radiomics workflow. All the image data were de-identified by assigning unique code numbers. The image files were loaded onto the investigator’s workstation. All the segmented ROIs from the T_1_-weighted images were mapped to co-registered IVIM images, and radiomic features were extracted using the PyRadiomics extension in 3D Slicer (version 5.6.1; http://www.slicer.org, accessed on 15 July 2025) [38]. Preprocessing for radiomic analysis involved applying a Gaussian filter, correcting for bias fields (N4ITK), resampling to 1 mm^3^ isotropic voxels, and image intensity normalisation.

Because IVIM maps are inherently quantitative, no intensity discretisation was performed; however, fixed bin widths supported texture analysis. Features were extracted for each lesion across *D*, *D**, and *f* maps, including shape descriptors (size and geometry), first-order statistics (intensity distribution), and texture features (grey-level spatial patterns derived from a Grey-Level Co-Occurrence Matrix (GLCM) and related matrices). Additional features were obtained from wavelet- and Gaussian-transformed images to capture multi-scale structural variations. The feature extraction adhered to the standards set by the Image Biomarker Standardisation Initiative (IBSI) [39].

To evaluate robustness, an inter-class correlation coefficient (ICC) analysis was conducted on 30 lesions to assess the reproducibility of radiomic features between observers. Only features with an ICC greater than 0.8, indicating high reproducibility, were retained for further analysis. All the retained features are considered reliable. To reduce dimensionality, mutual information feature selection and Principal Component Analysis (PCA) were employed [40,41]. PCA was also tested as an optional step for dimensionality reduction, with the number of components optimised as a hyperparameter [42]. The full, IBSI-compliant PyRadiomics parameterization—including preprocessing, discretization (per-map bin widths), texture-matrix settings, and filters—for each IVIM map (*f*, *D*, *D*) is provided in Appendix A.

### 2.5. Machine Learning-Based Predictive Modelling

#### 2.5.1. Model Training

The classification task involved lesion-level phenotyping, including enhancing and non-enhancing, as well as predicting patient-level clinical outcomes. Patients were categorised using five predefined binary thresholds: EDSS scores greater than 3.0 indicating disability, scores less than 4.0 indicating non-disability, documented use of walking aids for mobility, relapse activity within 1–3 years, and high-efficacy DMT usage for treatment status [21,43,44,45,46].

We evaluated multiple machine learning models, including two tree-based ensemble methods (eXtreme Gradient Boosting, XGBoost, and Random Forest, RF) [47,48] and two deep learning architectures (a residual neural network, RNN, and a convolutional neural network, CNN) [49,50]. Hyperparameter tuning was conducted using grid search within cross-validation [51,52], and the models’ effectiveness was thoroughly evaluated through fivefold cross-validation [52]. To address class imbalance, the Synthetic Minority Oversampling Technique (SMOTE) was applied to the training data. Dimensionality reduction in the radiomic feature space was performed using mutual information-based feature selection (selecting the top features between 1 and 20) [53], and optionally, Principal Component Analysis (PCA), with the number of components treated as a hyperparameter [54].

#### 2.5.2. Model Evaluation and Statistical Analysis

Data were divided into an 80% training and 20% testing split for both lesion-level and patient-level analyses, with stratification by patient to ensure no overlap between training and test subjects. Fivefold cross-validation on the training set was used to fine-tune model parameters. Class imbalance was addressed using SMOTE applied only to the training folds of a nested, patient-grouped cross-validation; no synthetic samples were included in validation or test sets. We performed within-fold feature reduction (mutual information/mRMR (Minimum Redundancy Maximum Relevance) with correlation pruning) and tuned hyperparameters in the inner loop. Sensitivity analyses compared SMOTE with class-weighted learners and focal-loss variants. The final model performance was assessed on the independent test set using accuracy and area under the curve (AUC) as primary metrics. We recorded the best cross-validation accuracy, the test accuracy, and the AUC with 95% confidence intervals (bootstrapped). All the analyses were implemented in Python (v3.10), and the performance metrics for each model (XGBoost, RF, RNN, and CNN) were tabulated for the different IVIM inputs (*f*, *D*, *D**, and the combined *f* + *D* + *D** feature set). The statistical significance for performance differences was set at *p* < 0.05, and AUC values were considered clinically relevant when the lower bound of the 95% confidence interval exceeded 0.70.

## 3. Results

### 3.1. Lesion Phenotyping (Enhancing vs. Non-Enhancing Lesions)

Radiomic features derived from IVIM parametric maps were highly effective in classifying enhancing versus non-enhancing lesions that encompassed both stable T_2_-hyperintense plaques and chronic T_1_-hypointense black holes. The most influential features for lesion phenotyping included several shape descriptors (e.g., 2D minor axis length, flatness) alongside texture metrics (entropy, grey-level non-uniformity, and run-length emphasis). These top discriminative features from each IVIM map are summarised in Figure 3.

Among the evaluated machine learning models, Random Forest achieved the highest performance using features from the IVIM-*f* and IVIM-*D** maps, with test accuracies of 96% and 91%, respectively, and AUCs of 0.99 in both cases. RNN also performed well on the *D* features, attaining an 88% test accuracy with an AUC of 0.99, while CNN achieved similarly strong results, particularly on the IVIM-*f* map (92% test accuracy, AUC 0.97). In contrast, XGBoost showed comparatively lower performance across the IVIM parameters, with test accuracies ranging from 65% to 83% and AUC values between 0.73 and 0.89, indicating reduced effectiveness in modelling these features.

The CNN model yielded similarly strong results on the IVIM-*f* map, achieving a ~92% test accuracy (AUC ≈ 0.97), nearly matching the Random Forest on this input (Table 2). In contrast, the CNN’s performance on IVIM-*D* was substantially lower (test accuracy ~53%, AUC ~0.56), reflecting the overall trend that models based on IVIM-*D* features were less predictive. For instance, Random Forest achieved only 71% test accuracy (AUC 0.73) with IVIM-*D*, and XGBoost models ranged from ~64.5% to 77% accuracy across the single-parameter IVIM maps (AUC 0.73–0.89).

Table 2 provides detailed performance metrics for each model, highlighting the superior discriminative ability of the perfusion-related IVIM parameters (*f* and *D**) compared to the pure diffusion parameter (*D*) in lesion activity classification.

### 3.2. Clinical Disability Prediction

#### 3.2.1. Expanded Disability Status Scale (EDSS)

The most radiomic features predictive of EDSS-based disability status were primarily lesion shape attributes (e.g., maximum 2D diameter, flatness, volume) and high-order texture metrics capturing lesion heterogeneity (e.g., grey-level variance, run-length non-uniformity). These features were repeatedly selected across different models. Notably, features extracted from the IVIM-*D* and IVIM-*D** maps were generally more informative for EDSS classification than those from IVIM-*f*, consistent with the idea that true diffusion and pseudo-diffusion characteristics of lesions correlate with disability. Figure 4 illustrates the top 20 ranked features from each IVIM map used in the models.

For EDSS classification, models trained on IVIM-*D* or IVIM-*D** features consistently outperformed those using IVIM-*f*. Random Forest using IVIM-*D* achieved the highest test accuracy at 89% (AUC = 0.90), indicating that lesions’ true diffusion metrics were very predictive of moderate-to-severe disability. The CNN models performed comparably well, achieving up to ~90% test accuracy on IVIM-*f* features (AUC ≈ 0.95) and about 88% on IVIM-*D* (AUC ≈ 0.91) (Table 3). The RNN model using IVIM-*D** features followed closely with 88% test accuracy and an AUC of 0.95. In contrast, XGBoost yielded more moderate performance (test accuracies from ~72% on IVIM-*D* to 83% on IVIM-*f*, with AUCs of 0.75–0.82 across the maps).

We also evaluated a combined-feature approach that incorporated radiomic features from all three IVIM maps (*f*, *D*, and *D** together) as input. This multi-parametric input yielded mixed results. For example, the RNN with combined IVIM features reached 89% test accuracy (AUC 0.93), roughly equivalent to its best single-map result, while the CNN’s combined-features model achieved about 80% test accuracy (AUC ~ 0.87), which was lower than its top performance on a single map. These findings suggest that, for EDSS prediction, the most informative diffusion metric (IVIM-*D* or IVIM-*D**) largely dominated the performance, and adding features from the other maps provided little incremental benefit in some models. Detailed performance metrics for all models and IVIM inputs are provided in Table 3.

#### 3.2.2. Mobility Impairment

The lesion features most associated with mobility disability mirrored those found in EDSS prediction. Lesions with greater size, more elongated shapes, and higher texture heterogeneity were more common in patients with impaired mobility. Figure 5 illustrates the top 20 features ranked from each IVIM map used to train the machine learning models.

Mobility impairment prediction yielded the best overall model performance among all clinical tasks. The RNN achieved the highest performance across all IVIM parameter maps, with test accuracies of 95.6% (IVIM-*f*), 91.2% (IVIM-*D*), and 92.6% (IVIM-*D**), and corresponding AUCs of 0.992, 0.979, and 0.987, respectively. Random Forest also performed well, particularly with IVIM-*D* (AUC = 0.958, test accuracy = 88.2%). XGBoost models demonstrated modest accuracy in this task, achieving 75–83.8% test accuracy and AUCs ranging from 0.836 to 0.866 across maps. Detailed metrics for these models are provided in Table 4.

### 3.3. Other Clinical Outcomes

Prediction of additional clinical variables, including relapse history, disease duration, and DMT status, was less successful. The best-performing relapse model (XGBoost with IVIM-*D*) reached approximately 75% test accuracy and AUC ~0.75. Disease duration models showed low classification accuracy (25–37%), approximating chance levels, while DMT status models performed below 65% accuracy with AUC values under 0.7. These findings suggest that lesion-level radiomic features may not sufficiently capture the complexity of treatment status or disease chronicity.

## 4. Discussion

Our findings demonstrate that radiomic analysis of IVIM-derived MRI features provides powerful insight into multiple sclerosis lesion activity and disability. At the lesion level, radiomic models accurately distinguished gadolinium-enhancing lesions from non-enhancing lesions using only non-contrast IVIM maps. Notably, features from the IVIM-*f* map yielded near-perfect classification performance (test AUC ~ 0.99 using Random Forest in our study), indicating that active inflammatory lesions exhibit unique microvascular characteristics detectable without contrast. This aligns with quantitative IVIM analyses showing that acutely enhancing lesions have significantly different diffusion and perfusion properties compared to chronic lesions [17]. For example, gadolinium-enhancing lesions have lower ADC and pure diffusion IVIM-*D* values than non-enhancing lesions but higher IVIM-*f* values, reflecting increased microcirculatory perfusion [55]. These IVIM-based differences likely arise from the cellular inflammation and blood–brain barrier disruption in active plaques versus the greater extracellular water diffusion in chronic lesions with tissue loss [56,57].

Importantly, our IVIM radiomic features were also strongly associated with patient-level clinical outcomes. We observed that models incorporating lesion radiomics could predict disability status as measured by EDSS and mobility impairment with high accuracy (cross-validated AUC ~ 0.90). In practical terms, patients with a greater lesion burden of certain radiomic signatures (e.g., lesions with a more irregular shape or heterogeneous texture) tended to have higher disability, and this effect was particularly evident for non-enhancing lesions, which appeared more strongly associated with disability and mobility outcomes than enhancing lesions or T_1_ black holes [56]. This suggests that subtle imaging markers of microstructure captured by radiomics may bridge the classic “clinico-radiologic paradox”, wherein conventional MRI lesion load explains only a fraction of MS disability [58].

Our approach, leveraging advanced diffusion/perfusion features, appears to narrow that gap by capturing lesion qualities linked to functional impairment. For instance, among the top predictive features in our models were lesion shape descriptors (e.g., minor axis length) and texture metrics (e.g., grey-level dependence variance), implying that larger, flatter lesions with complex intensity patterns are associated with worse clinical status [59]. Such features likely reflect the presence of extensive tissue damage or prolonged inflammation in lesions, which in turn manifest as greater neurological deficits [59]. Overall, the strong performance of the radiomic models in classifying lesion activity and estimating disability underlines the potential of non-contrast IVIM radiomics as a comprehensive imaging biomarker for MS disease state.

Our results are consistent with and extend prior studies in several ways. The IVIM-based differences we observed between active and chronic lesions concur with recent quantitative MRI findings. Alomair et al. [17] reported that enhancing MS lesions show significantly lower ADC and IVIM-*D* values compared to non-enhancing lesions (indicating more restricted water diffusion in acute inflammation) as well as higher perfusion fraction values consistent with increased lesion vascularity [17].

These physiological differences form the basis for our radiomics-driven classification. By extracting texture and morphology features from IVIM parametric maps, we capitalised on the same underlying contrasts noted in quantitative analyses. To the best of our knowledge, our study is the first to apply radiomics on IVIM MRI in MS, and the near-perfect lesion classification we achieved (AUC ~ 0.98) slightly exceeds what has been reported with conventional MRI-based approaches. For example, Rostami et al. [24] used radiomic features from standard T_2_-FLAIR images to distinguish active (Gd-enhancing) from inactive lesions, and their best machine learning model (an ensemble gradient boosting classifier) achieved an AUC of ~0.87 on the test set [24]. A deep learning model in their study performed better (AUC ~ 96%), highlighting the promise of advanced algorithms; nonetheless, our IVIM radiomics approach, using only non-contrast diffusion/perfusion maps, reached comparable or higher discriminative performance. Similarly, Narayana et al. [60] explored deep learning to predict lesion enhancement from multi-sequence unenhanced MRI. Their CNN (using pre-contrast T_1_, T_2_, and FLAIR inputs) obtained a sensitivity of 78% and specificity of 73% in identifying enhancing lesions [60].

Our single-time study also produced a higher prediction performance compared to that of a conventional MRI used in a follow-up study (AUC of 0.857) [61]. This highlights the potential for our approach to be used for predicting the type of MS lesion in follow-up studies. Our radiomics-based method achieved higher specificity and sensitivity compared to radiomics using only anatomical images, likely because IVIM maps provide direct surrogates of lesion perfusion that conventional structural MRI lacks. These comparisons suggest that incorporating tissue microdynamic information via IVIM can boost the accuracy of non-contrast lesion activity detection.

In terms of disability prediction, few prior works have attempted to link radiomic features with clinical status in MS. A recent study by Pontillo et al. [58] combined radiomic, volumetric, and connectivity metrics from routine MRIs to predict EDSS, reporting a correlation of r ≈ 0.8 between the model’s output and actual disability [58]. Our results are in line with this magnitude of association, reinforcing that high-dimensional MRI features can capture meaningful information about neurological impairment. Notably, the most informative features in Pontillo et al.’s model [58] included lesion distribution and texture measures in specific brain regions, paralleling our finding that radiomic descriptors of lesion morphology/heterogeneity carry prognostic value. On the other hand, some investigators have raised caution about the incremental benefit of complex radiomics over simpler MRI markers. For example, one analysis found that adding radiomic features from the baseline MRI did not significantly improve 10-year disability predictions compared to conventional lesion and volume measures alone [28]. This shows that while our cross-sectional models exhibit excellent performance, radiomics is not a panacea and must prove its robustness and added value beyond standard metrics in longitudinal settings.

Nonetheless, our study breaks new ground by demonstrating that IVIM-based radiomics is a novel, physiology-oriented imaging approach that can successfully classify lesion activity and correlate with clinical outcomes. This complements earlier efforts using structural MRI or PET (positron emission tomography), and it pushes the envelope by suggesting that non-contrast diffusion MRI can serve dual roles in MS: identifying active lesions and gauging disease severity.

A study that combined PET and MRI with automatic lesion segmentation had been shown to be powerful in predicting the disease annual relapse rate (AUC of 0.96) [62]. Our approach used a single non-ionising MR imaging technique, which is more commonly used in neurological assessment for MS patients, and radiomics to predict EDSS. However, the utility of our approach to predict disease relapses has not been measured; this will be addressed in the future.

If validated in broader cohorts, the ability to characterise lesions and predict disability using IVIM radiomic features could have significant clinical ramifications. Foremost, it offers a pathway to reduce or even eliminate the routine use of gadolinium contrast in MS MRI monitoring. Gadolinium-enhanced T_1_ scans are currently the gold standard for detecting active lesions, but repeated gadolinium exposure has well-known downsides—including rare but serious adverse effects such as nephrogenic systemic fibrosis and evidence of gadolinium deposition in the brain and body tissues [14,15].

Our results suggest that an IVIM MRI, combined with machine learning analysis, can identify active inflammatory lesions with high fidelity without the need for contrast injection. In practical terms, this could make serial MS imaging safer for individuals who require frequent follow-up scans. An AI-driven tool that flags lesions likely to enhance from non-contrast scans could be incorporated into the radiology workflow, alerting clinicians to new disease activity in real time and prompting timely therapeutic interventions, all while avoiding unnecessary contrast administration.

Beyond lesion detection, the strong correlation between radiomic features and clinical disability suggests that MRI could help guide patient management. While EDSS and mobility impairment are typically assessed clinically, radiomics may provide imaging biomarkers that reflect disease burden more accurately than lesion count alone. Patients with radiomic signatures linked to higher EDSS could be flagged early for closer monitoring or intensified therapy, even if conventional imaging appears unremarkable. These features may capture subtle pathology, such as microstructural damage or abnormal perfusion not visible to the human eye, supporting more personalised treatment decisions. Notably, our models also predicted treatment status, implying that radiomics may detect the cumulative effects of disease-modifying therapies. This raises the potential for radiomics to contribute to ongoing therapy monitoring, where changes in lesion profiles could signal emerging disease activity.

This study has several limitations that temper the interpretation of our results. First, our analysis was cross-sectional, evaluating lesion characteristics and clinical status at a single time point. This limits our ability to assess whether IVIM radiomic features can predict future disease activity or disability progression. Longitudinal studies are needed to determine if the models can prognosticate outcomes such as subsequent relapse rate or EDSS worsening over time.

Second, our cohort was drawn from a single centre using a specific 1.5 T MRI scanner and IVIM protocol. A strength of 1.5 Tesla is the minimum for MS diagnosis according to the McDonald criteria [63]. However, a higher magnetic field, such as 3 T, generally provides a higher SNR and consequently higher spatial resolution, improving lesion conspicuity [64,65]. Radiomics feature distributions may differ between 1.5 T and 3 T, meaning that features optimised at 1.5 T cannot always be directly applied at 3 T. Therefore, further investigations using higher field strengths are warranted [66]. While we took care to standardise imaging and ROI segmentation, the generalisability of our radiomic models to other centres, scanners, or field strengths is unproven. Multi-centre validations will be essential to ensure the robustness of these features and models across diverse imaging settings.

Third, the radiomic approach is inherently complex and dependent on multiple preprocessing steps (registration, lesion segmentation, and feature extraction) [67]. Any variability in these steps, for example, slight differences in how lesions are outlined, could affect the features. We mitigated this by having experienced neuroradiologists define lesions and using consistent software, but some degree of inter-rater variability is unavoidable. Future work might incorporate automated lesion segmentation algorithms to standardise ROI definition (though these algorithms must themselves be accurate for IVIM images). While SMOTE mitigates class imbalance, it does not increase the number of independent patients or replace larger cohorts; consequently, sample size remains a limitation. Future multi-centre, longitudinal studies with external validation are required to confirm generalisability and to assess delta-radiomics.

Finally, while we focused on lesion-level features, MS pathology is not confined to focal lesions; diffuse normal-appearing white matter damage [68,69] and cortical pathology [70] also drive disability. Our radiomics approach did not capture these aspects, which could partly explain why the model predictions, although strong, are not perfect. Integrating radiomic features from normal-appearing tissue or other sequences (e.g., volumetric atrophy measures [71], cortical lesions on Double Inversion Recovery images [72,73]) might further improve the predictive power.

## 5. Conclusions

In conclusion, we showed that IVIM radiomics provided important features that can be utilised for characterising MS lesions and predicting patient disability with high accuracy. A combination of IVIM radiomics and machine learning was highly predictive of the EDSS and mobility status in patients with relapsing–remitting MS.

## Figures and Tables

**Figure 1 jcm-14-06753-f001:**
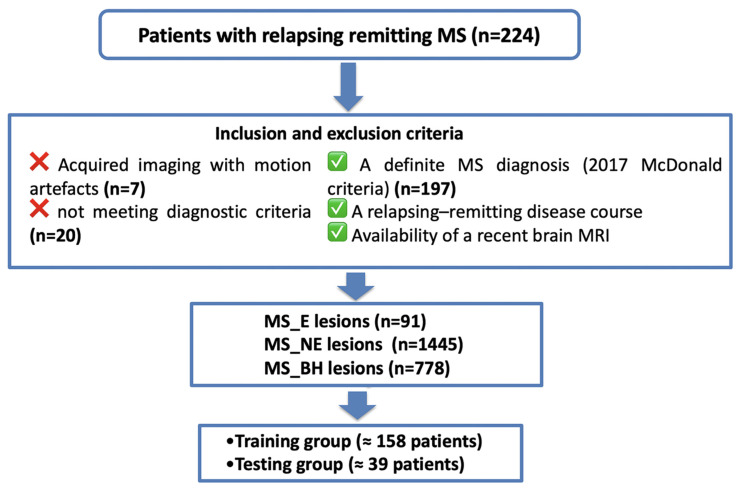
Flowchart of patient’s selection criteria and processing of data.

**Figure 2 jcm-14-06753-f002:**
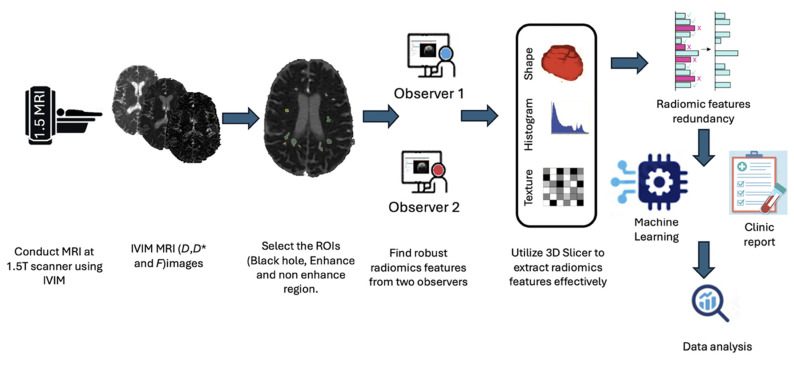
Schematic diagram of the radiomic analysis pipeline’s steps.

**Figure 3 jcm-14-06753-f003:**
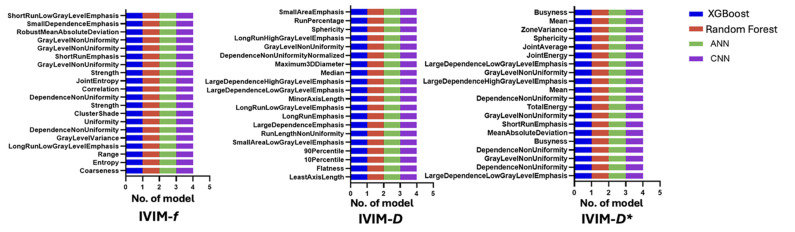
The top 20 most important IVIM radiomic features to distinguish between enhanced and non-enhanced lesions.

**Figure 4 jcm-14-06753-f004:**
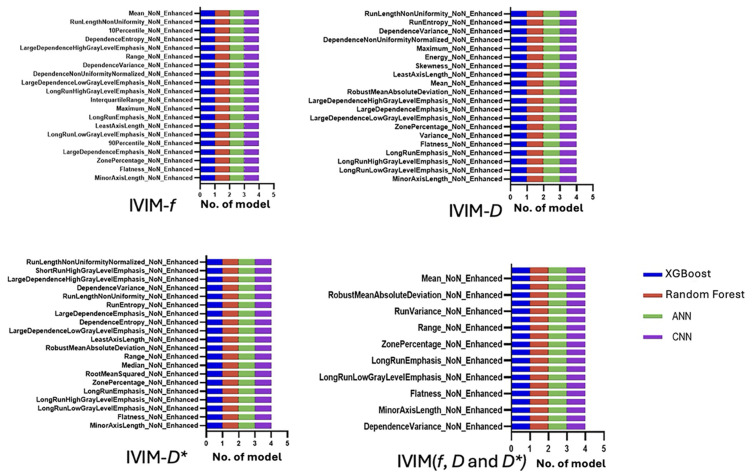
The top 20 most important radiomic features to differentiate between EDSS groups.

**Figure 5 jcm-14-06753-f005:**
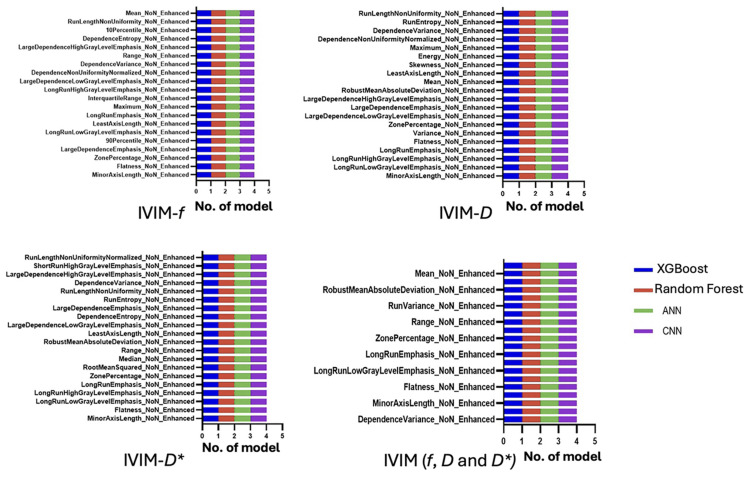
The top 20 most important radiomic features to differentiate between mobility groups.

**Table 1 jcm-14-06753-t001:** Demographic and clinical data of RR-MS patients.

Demographic and Clinical Data	Outcome Mean ± SD and %
Mean age ± SD	36.1 ± 9.4 years
Gender (male and female)	Male = 59 (29.9%) Female = 138 (70.1%)
Disease duration	6.3 ± 5.2 years
EDSS (mean ± SD)	2.25 ± 1.91
N relapses (mean ± SD)	463 for all RR-MS patients 2.65 ± 2.03
N patients with DMT	144 (73.1%)
N patients without DMT	53 (26.9%)
N patients with mobility impairment	20 (10.2%)
N patients with normal walking ability	177 (89.8%)
N MS_E lesions (mean ± SD)	91 (0.46 ± 1.65)
N MS_NE lesions (mean ± SD)	1445 (7.34 ± 4.59)
N MS_BH lesions (mean ± SD)	778 (3.95 ± 4.95)

Note: RR-MS = relapsing–remitting multiple sclerosis; N = number; SD = standard deviation; EDSS = Expanded Disability Status Scale; DMT = disease-modifying treatment; MS_E = multiple sclerosis enhanced lesions; MS_NE = multiple sclerosis non-enhanced lesions; MS_BH = multiple sclerosis black hole lesions.

**Table 2 jcm-14-06753-t002:** Model performance for classifying enhancing vs. non-enhancing MS lesions from IVIM radiomics features.

Model	MRI	Train Accuracy	Test Accuracy	AUC	95% AUC Confidence Interval (CI)
XGBoost	IVIM-*f*	0.84	0.77	0.89	0.82–0.94
	IVIM-*D*	0.79	0.645	0.73	0.6–0.85
	IVIM-*D**	0.9	0.83	0.88	0.81–0.94
Random Forest	IVIM-*f*	0.98	0.96	0.99	0.93–1
	IVIM-*D*	0.98	0.71	0.73	0.72–0.93
	IVIM-*D**	0.98	0.91	0.99	0.96–1
RNN	IVIM-*f*	0.95	0.89	0.93	0.84–0.94
	IVIM-*D*	0.71	0.65	0.74	0.54–0.75
	IVIM-*D**	0.95	0.88	0.99	0.88–0.96
CNN	IVIM-*f*	0.96	0.92	0.97	0.93–0.99
	IVIM-*D*	0.61	0.53	0.56	0.41–0.71
	IVIM-*D**	0.96	0.89	0.96	0.91–0.99

Note: XGBoost = eXtreme Gradient Boosting; RNN = residual neural network; CNN = convolutional neural network; AUC = area under the curve; *f* = perfusion fraction; *D* = diffusion coefficient; *D** = pseudo-diffusion coefficient; the units for *D* and *D** are 10^−3^ mm^2^/s.

**Table 3 jcm-14-06753-t003:** Evaluation of machine learning models for classification of IVIM parameters with EDSS.

Model	MRI	Train Accuracy	Test Accuracy	AUC	95% AUC Confidence Interval (CI)
XGBoost	IVIM-*f*	0.78	0.83	0.82	0.71–0.92
	IVIM-*D*	0.8	0.72	0.75	0.63–0.87
	IVIM-*D**	0.85	0.81	0.99	0.79–0.96
	IVIM (*f*, *D*, and *D**)	0.8	0.81	0.88	0.78–0.96
Random Forest	IVIM-*f*	0.94	0.84	0.87	0.76–0.96
	IVIM-*D*	0.95	0.89	0.9	0.79–0.98
	IVIM-*D**	0.95	0.8	0.83	0.72–0.93
	IVIM (*f*, *D*, and *D**)	0.98	0.81	0.84	0.73–0.93
RNN	IVIM-*f*	0.97	0.84	0.93	0.75–0.93
	IVIM-*D*	0.97	0.83	0.92	0.73–0.92
	IVIM-*D**	0.96	0.88	0.95	0.78–0.95
	IVIM (*f*, *D*, and *D**)	0.97	0.89	0.93	0.82–0.96
CNN	IVIM-*f*	0.97	0.9	0.947	0.88–0.99
	IVIM-*D*	0.98	0.88	0.91	0.74–0.95
	IVIM-*D**	0.97	0.8	0.90	0.82–0.97
	IVIM (*f*, *D*, and *D**)	0.96	0.8	0.87	0.78–0.95

Note: XGBoost = eXtreme Gradient Boosting; RNN = residual neural network; CNN = convolutional neural network; AUC = area under the curve; *f* = perfusion fraction; *D* = diffusion coefficient; *D** = pseudo-diffusion coefficient; the units for *D* and *D** are 10^−3^ mm^2^/s.

**Table 4 jcm-14-06753-t004:** Evaluation of machine learning models for classification of IVIM parameters with mobility assessment.

Model	MRI	Train Accuracy	Test Accuracy	AUC	95% AUC Confidence Interval (CI)
XGBoost	IVIM-*f*	0.82	0.75	0.84	0.72–0.93
	IVIM-*D*	0.86	0.84	0.85	0.74–0.94
	IVIM-*D**	0.83	0.79	0.87	0.76–0.96
	IVIM (*f*, *D*, and *D**)	0.81	0.84	0.90	0.81–0.98
Random Forest	IVIM-*f*	0.97	0.87	0.95	0.88–0.99
	IVIM-*D*	0.95	0.88	0.96	0.9–0.99
	IVIM-*D**	0.96	0.91	0.95	0.89–0.99
	IVIM (*f*, *D* and *D**)	0.96	0.90	0.93	0.86–0.99
RNN	IVIM-*f*	0.97	0.96	0.99	0.91–1.00
	IVIM-*D*	0.93	0.91	0.98	0.84–0.97
	IVIM-*D**	0.95	0.93	0.99	0.87–0.98
	IVIM (*f*, *D* and *D**)	0.99	0.97	0.99	0.93–1.00
CNN	IVIM-*f*	0.97	0.96	0.98	0.95–1.00
	IVIM-*D*	0.97	0.97	0.99	0.98–1.00
	IVIM-*D**	0.97	0.86	0.96	0.932–1.00
	IVIM (*f*, *D* and *D**)	0.92	0.92	0.95	0.89–0.99

Note: XGBoost = eXtreme Gradient Boosting; RNN = residual neural network; CNN = convolutional neural network; AUC = area under the curve; *f* = perfusion fraction; *D* = diffusion coefficient; *D** = pseudo-diffusion coefficient; the units for *D* and *D** are 10^−3^ mm^2^/s.

## Data Availability

The data presented in this study are available on request from the corresponding author, as they are currently being used in another clinical experiment.

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
