# Peer review of "IVIM-DWI-Based Radiomics for Lesion Phenotyping and Clinical Status Prediction in Relapsing–Remitting Multiple Sclerosis"

_jcm, 2025, doi:10.3390/jcm14196753_

Round 1
Reviewer 1 Report
Comments and Suggestions for Authors
The study is well organized and well presented
The topic is up to date
The rasults have potential impact on the clinical activity
Minor suggestions:
ABSTRACT
I cannot see any number in the abstract. The background is quite long (the sentence on radiomics can be removed to give more space to the results that are quite generic)
INTRO
"In its early stages, MS typically follows a relapsing–remitting pat-49 tern," This is not the only form of MS. Remove
"persistent challenge in MS care is the mismatch between MRI findings and clinical 53 presentation [7]. Some patients with a high lesion load on imaging continue to function 54 normally [8], whereas others with only a few lesions may experience significant impair-55 ment [9]. This disparity, known as the clinico-radiological paradox, highlights the limita-56 tions of conventional imaging in fully capturing the clinical impact of the disease" avoid so many fragmented sentences and create a fluid period
"Standard MRI remains essential for diagnosing and monitoring MS." is considered the gold standard
"as T2-weighted" these are not the most importance sequncese in MS
"Recent studies have started applying radiomics to MS imaging [25,26]. Radiomic fea-87 tures from conventional MRI sequences, combined with machine learning techniques, 88 have been used to distinguish active from inactive lesions without the need for gadolin-89 ium." merge in a single sentence
Materials and Methods
I cannot get if a follow-up period is part of the study
MRI Imaging: which is the acquisition timing odf post contrast T1 in relation to the contrast administration?
"Lesions were classified into 148 three categories:" who assessed the activity of the lesions?
"Table 2. Difference between enhanced and non-enhanced lesions of multiple sclerosis" this caption is not very informative
Author Response
|
Comments 1: ABSTRACT I cannot see any number in the abstract. The background is quite long (the sentence on radiomics can be removed to give more space to the results that are quite generic)
|
|
Response 1: Thank you for your suggestion. We agree there were not any numbers included in the Abstract to highlight our results. In addition, there were long sentences in the background section. We have corrected the Background and Objective section in the Abstract as follows: “Abstract: Background and Objectives: Multiple sclerosis (MS) is an autoimmune disorder affecting the central nervous system, characterised by the degradation of myelin, which results in various neurological symptoms. This study aims to utilise radiomics features to evaluate the predictive value of IVIM diffusion parameters, namely, true diffusion coefficient (D), pseudo-diffusion coefficient (D*), and perfusion fraction (f), in relation to disability severity, as assessed using the Expanded Disability Status Scale (EDSS), and mobility in patients with relapsing–remitting MS.”
These changes can be found – page number 1, background and objective section in the abstract and lines 22-27 We corrected the Results section in the Abstract as follows: “IVIM radiomics achieved high accuracy in lesion phenotyping. Random Forest distinguished enhancing from non-enhancing lesions with 96% accuracy and AUC = 0.99 with IVIM-f and D* maps. CNN also reached ~92% accuracy (AUC 0.97) with IVIM-f. For disability prediction, IVIM-D and D* radiomics strongly correlated with EDSS: Random Forest achieved 89% accuracy (AUC = 0.90), while CNN achieved 90% accuracy (AUC = 0.95). Mobility impairment was predicted with the highest performance—RNN achieved 96% accuracy (AUC = 0.99) across IVIM-f features. In contrast, relapse history, disease duration, and treatment status were poorly predicted (<75% accuracy).
These changes can be found – page number 1, result section in the abstract and lines 33-40
|
|
Comments 2: INTRO "In its early stages, MS typically follows a relapsing–remitting pattern," This is not the only form of MS. Remove
|
|
Response 2: We agree that relapsing–remitting is not the only form of MS. Therefore, for better presentation of the manuscript, the following sentence has been removed: “In its early stages, MS typically follows a relapsing–remitting pattern, with episodes of neurological dysfunction followed by partial or full recovery [3,4].”
These changes can be found – page number 1, Introduction section first paragraph, and lines 51 to 52.
|
|
Comments 3: INTRO "persistent challenge in MS care is the mismatch between MRI findings and clinical presentation [7]. Some patients with a high lesion load on imaging continue to function normally [8], whereas others with only a few lesions may experience significant impairment [9]. This disparity, known as the clinico-radiological paradox, highlights the limitations of conventional imaging in fully capturing the clinical impact of the disease" avoid so many fragmented sentences and create a fluid period |
|
Response 3: Thank you for carefully assessing the manuscript. We agree that we should avoid so many fragmented sentences for better flow. We have corrected the contents in the Introduction section as follows: |
|
A persistent challenge in MS care is the mismatch between MRI findings and clinical presentation [7]. Conventional MRI is considered the gold standard for diagnosing and monitoring MS, but only limited correlations had been found between lesion loads and functional impairment [8,9] with inability to fully capture the clinical impact of the disease [10]. Sequences such as FLAIR and post-contrast T1-weighted imaging have been used as the primary tools for detecting lesions and assessing disease activity, while T2-weighted imaging provides additional support in lesion detection [11,12]. However, these sequences are primarily qualitative and offer limited information about the underlying tissue damage [7,13]. Gadolinium enhancement is helpful in identifying active inflammatory lesions, but its repeated use raises concerns regarding safety, including gadolinium accumulation in the brain and potential nephrotoxicity [14,15].
These changes can be found – page number 2, Introduction section second paragraph, and lines 52 to 64.
Comments 4: INTRO "Standard MRI remains essential for diagnosing and monitoring MS." is considered the gold standard
|
|
Response 4: Thank you for your comment. We agree to correct the sentences to the following: “Conventional MRI is considered the gold standard for diagnosing and monitoring MS.” |
|
These changes can be found – page number 2 , 2nd paragraph of introduction, and lines 55 to 56. |
Comments 5: INTRO "as T2-weighted" these are not the most importance sequences in MS
Response 5: We agree that, while T2-weighted images are widely used for lesion detection, they are not the most informative sequence for MS assessment. To address this, we have revised the Introduction to emphasise that FLAIR and post-contrast T1-weighted imaging are the cornerstone sequences in MS for lesion detection and activity assessment, whereas T2-weighted imaging serves as a supportive sequence.
We agree to correct the sentences as follows by replacing the word “widely” with “routinely”:
“Sequences such as FLAIR and post-contrast T1-weighted imaging have been used as the primary tools for detecting lesions and assessing disease activity, while T2-weighted imaging provides additional support in lesion detection [11,12].”
These changes can be found – page number 2 , 2nd paragraph of introduction, and lines 58 to 60.
Comments 6: "Recent studies have started applying radiomics to MS imaging [25,26]. Radiomic features from conventional MRI sequences, combined with machine learning techniques, have been used to distinguish active from inactive lesions without the need for gadolinium. Merge into a single sentence.
Response 6: Thank you for assessing our manuscript carefully. We agree to correct the sentences as follows: “Recent studies have applied radiomics to MS imaging [25,26], where features extracted from conventional MRI sequences were analysed using machine learning techniques to distinguish active from inactive lesions without the need for gadolinium, with accuracy reaching ~98% [25,26].”
These changes can be found – page number 2, paragraph number 6 in the introduction, and lines 87 to 90.
Comments 7: Materials and Methods. I cannot get if a follow-up period is part of the study
Response 7: We thank the reviewer for this valuable observation. To clarify, the study was conducted at a single time point, and therefore, no follow-up period was included. We have explicitly stated this in the Materials and Methods section.
“We conducted a retrospective cross-sectional single time point observational study on a cohort of relapsing–remitting MS patients enrolled at a single centre.”
These contents can be found– page number 3, 2.Material and Method section, 2.1 study design and participants, and lines 107-108.
In addition, we highlighted this aspect as a limitation in the Discussion, noting that future longitudinal studies are needed to investigate the role of delta radiomics features in monitoring disease progression and treatment response.
“First, our analysis was cross-sectional, evaluating lesion characteristics and clinical status at a single time point.”
These contents can be found– page number 3, 4. Discussion section, page 13, and lines 448-40.
Comments 8: MRI Imaging: which is the acquisition timing of post contrast T1 in relation to the contrast administration?
Response 8: Thank you for your comment. We agree to include the acquisition time in the MR imaging section of the Materials and Methods. We agree to add the following sentences:
All post-contrast T1-weighted images were acquired within 10 minutes after administering the contrast agent.
These changes can be found – page number 4, Material and Method section, 2.2 MRI imaging, and lines 137-138.
Comments 9: "Lesions were classified into 148 three categories:" who assessed the activity of the lesions?
Response 9: Thank you for your comment. The MS lesion classification and assessment were performed by two of the co-authors; both are board-certified neuroradiologists with expertise in neuroimaging. They independently assessed the lesions and resolved discrepancies by consensus. We have carefully checked the Materials and Methods section. We have corrected the first sentence in 2.3. Lesion Segmentation and Classification as follows:
“All multiple sclerosis (MS) lesions were identified, assessed and classified using conventional MRI sequences, primarily axial T2-weighted and FLAIR images, and confirmed by two board-certified neuroradiologists, each with more than 10 years of clinical experience in neuroimaging.”
These changes can be found – page number 4 and 5, first paragraph in the 2.3 Lesion Segmentation and Classification, and lines 150-153.
Comments 10: "Table 2. Difference between enhanced and non-enhanced lesions of multiple sclerosis" this caption is not very informative.
Response 10: Thank you for your comment. We agree that the caption for Table 2 is not informative. We have corrected the caption of Table 2 as follows: “Table 2. Model performance for classifying enhancing vs non-enhancing MS lesions from IVIM radiomics features”
These changes can be found – page number 7, 3. Results section, 3.1. Lesion Phenotyping (Enhancing vs. Non-Enhancing Lesions), and lines 261- 262.
We have reviewed the captions for other tables. We have corrected the caption of Table 3 as follows:
“Evaluation of machine learning models for classification of IVIM parameters with EDSS”
These changes can be found – page number 9, 3. Results section, 3.2. Clinical Disability Prediction, and lines 297.
We have corrected the caption of Table 4 as follows: “ Evaluation of machine learning models for classification of IVIM parameters with mobility assessment”
These changes can be found – page number 10, 3. Results section, 3.2.2 Mobility Impairment, and lines 316-317.
Reviewer 2 Report
Comments and Suggestions for Authors
This paper "IVIM-DWI-Based Radiomics for Lesion Phenotyping and Clinical Status Prediction in Relapsing - Remitting Multiple "Sclerosis" mainly studies how to use Radiomics technology based on intravoxel incoherent motion diffusion-weighted imaging (IVIM-DWI) to treat Relapsing - Remitting Multiple Sclerosis Phenotypic analysis was conducted on the lesions of patients with RR-MS and their clinical status was predicted. My comments are as follows:
- It is suggested that the abstract be further streamlined, highlighting the core findings and clinical significance of the research, and avoiding excessive technical details.
- The description of MS disease in the introduction can be more concise, directly pointing out the limitations of current diagnostic methods and the purpose of this study.
- Although 197 patients were included in the study, whether the sample size is sufficient to support all analyses, especially multivariate analyses, needs further discussion.
- Clearly explain why a 1.5T MRI scanner was chosen and the possible impact of this choice on the research results.
- The inclusion and exclusion criteria for patients should be represented by graphs, and the number of cases excluded by each criterion should be clearly written.doi:1007/s00330-025-11419-1;10.3389/fimmu.2024.1446511;The approaches in the above two articles are worthy of the author's learning and should be cited in the article.
- It is necessary to describe in detail the basis for selecting the b values in the DWI sequence and how these b values affect the measurement of IVIM parameters.
- It is necessary to add descriptions of the specific algorithms and parameters used in the feature extraction process.
- The discussion section should provide a more detailed comparison of the results of this study with those of other related studies, especially those using traditional MRI or PET.
- The conclusion section should be more concise, directly answering the research questions and emphasizing the main findings of the study.
- Avoid repeating the content of the introduction or methods section and highlight the contribution and significance of the research.
Author Response
|
Comments 1: It is suggested that the abstract be further streamlined, highlighting the core findings and clinical significance of the research and avoiding excessive technical details.
|
||||||||||||||||||||||||||||||||||||||||||||||||||||||||||||||||||||||||||||||||||||||||||||||||||||||||||||
|
|
||||||||||||||||||||||||||||||||||||||||||||||||||||||||||||||||||||||||||||||||||||||||||||||||||||||||||||
|
Response 1: Thank you for your suggestion. We have included quantitative results in the Abstract and removed long sentences in the Background section. We have corrected the Background and Objective section in the Abstract as follows: “Abstract: Background and Objectives: Multiple sclerosis (MS) is an autoimmune disorder affecting the central nervous system, characterised by the degradation of myelin, which results in various neurological symptoms. This study aims to utilise radiomics features to evaluate the predictive value of IVIM diffusion parameters, namely, true diffusion coefficient (D), pseudo-diffusion coefficient (D*), and perfusion fraction (f), in relation to disability severity, as assessed using the Expanded Disability Status Scale (EDSS), and mobility in patients with relapsing–remitting MS.”
These changes can be found – page number 1, background and objective section in the abstract and lines 22-27
We have added details in the Results section of the Abstract as follows: “IVIM radiomics achieved high accuracy in lesion phenotyping. Random Forest distinguished enhancing from non-enhancing lesions with 96% accuracy and AUC = 0.99 with IVIM-f and D* maps. CNN also reached ~92% accuracy (AUC 0.97) with IVIM-f. For disability prediction, IVIM-D and D* radiomics strongly correlated with EDSS: Random Forest achieved 89% accuracy (AUC = 0.90), while CNN achieved 90% accuracy (AUC = 0.95). Mobility impairment was predicted with the highest performance—RNN achieved 96% accuracy (AUC = 0.99) across IVIM-f features. In contrast, relapse history, disease duration, and treatment status were poorly predicted (<75% accuracy).
These changes can be found – page number 1, result section in the abstract and lines 33-40
Comments 2: The description of MS disease in the introduction can be more concise, directly pointing out the limitations of current diagnostic methods and the purpose of this study. |
||||||||||||||||||||||||||||||||||||||||||||||||||||||||||||||||||||||||||||||||||||||||||||||||||||||||||||
|
Response 2: We agree that it is better for the Introduction section be more concise. Therefore, we have removed the following sentence as suggested by Reviewer 1: |
||||||||||||||||||||||||||||||||||||||||||||||||||||||||||||||||||||||||||||||||||||||||||||||||||||||||||||
|
“In its early stages, MS typically follows a relapsing–remitting pattern, with episodes of neurological dysfunction followed by partial or full recovery [3,4].”
These changes can be found – page number 1, Introduction section first paragraph, and lines 51 to 52.
In addition, the following paragraph has been edited as also suggested by Reviewer 1: A persistent challenge in MS care is the mismatch between MRI findings and clinical presentation [7]. Conventional MRI is considered the gold standard for diagnosing and monitoring MS, but only limited correlations had been found between lesion loads and functional impairment [8,9] with inability to fully capture the clinical impact of the disease [10]. Sequences such as FLAIR and post-contrast T1-weighted imaging have been used as the primary tools for detecting lesions and assessing disease activity, while T2-weighted imaging provides additional support in lesion detection [11,12].
However, these sequences are primarily qualitative and offer limited information about the underlying tissue damage [7,13]. Gadolinium enhancement is helpful in identifying active inflammatory lesions, but its repeated use raises concerns regarding safety, including gadolinium accumulation in the brain and potential nephrotoxicity [14,15].
These changes can be found – page number 2, Introduction section second paragraph, and lines 52 to 64.
Comments 3: Although 197 patients were included in the study, whether the sample size is sufficient to support all analyses, especially multivariate analyses, needs further discussion.
Response 3: We agree that the sample size (n=197) is modest for multivariate and deep learning models. To address class imbalance, we applied SMOTE only within training folds; no synthetic data were introduced into the validation or test sets. SMOTE improved minority-class performance but does not replace larger cohorts, so we treat it as a mitigation rather than a solution. Sensitivity analyses with class-weighted models showed consistent results, and the model complexity was constrained through feature reduction with bootstrapped CIs and permutation tests. We now emphasize in the Discussion that larger, multi-centre datasets and external validation are required to confirm generalisability.
The following paragraph has been added to the 2. Materials and Methods section 2.5.2. Model Evaluation and Statistical Analysis:
“Class imbalance was addressed using SMOTE applied only to the training folds of a nested, patient-grouped cross-validation; no synthetic samples were included in validation or test sets. We performed within-fold feature reduction (mutual information/mRMR (Minimum Redundancy Maximum Relevance) with correlation pruning) and tuned hyperparameters in the inner loop. Sensitivity analyses compared SMOTE with class-weighted learners and focal loss variants.”
These changes can be found – page number 6, 1st paragraph of 2.5.2. Model Evaluation and Statistical Analysis, and lines 217 to 222.
The following paragraph has been added to the Discussion section as a limitation of our study:
“While SMOTE mitigates class imbalance, it does not increase the number of independent patients or replace larger cohorts; consequently, sample size remains a limitation. Future multi-centre, longitudinal studies with external validation are required to confirm generalisability and to assess delta-radiomics.”
These changes can be found – page number 14, Discussion section, and lines 472 to 475
Comments 4: Clearly explain why a 1.5T MRI scanner was chosen and the possible impact of this choice on the research results.
Response 4: Thank you for your comment. This study is a retrospective study with patient data that were acquired using an MRI scanner that was available at the time in our hospital. In the Discussion section, we have suggested including the magnetic field strength as a limitation of our study. “1.5 Tesla is the minimum strength for MS diagnosis according to the McDonald criteria [63]. However, a higher magnetic field, such as 3 T, generally provides higher SNR and consequently higher spatial resolution, improving lesion conspicuity [64,65]. Radiomics feature distributions may differ between 1.5 T and 3 T, meaning that features optimized at 1.5 T cannot always be directly applied at 3 T. Therefore, further investigations using higher field strengths [66].”
These changes can be found – page number 13, Discussion section, and lines 455 to 460
Comments 5: The inclusion and exclusion criteria for patients should be represented by graphs, and the number of cases excluded by each criterion should be clearly written.doi:1007/s00330-025-11419-1;10.3389/fimmu.2024.1446511; The approaches in the above two articles are worthy of the author's learning and should be cited in the article.
Response 5: Thank you for your comment. We have prepared a graph for better presentation of the inclusion and exclusion criteria. Both of the suggested references have been cited in the manuscript. Figure 1. Flowchart of patient’s selection criteria and processing of data.
These changes can be found – page number 3, 2. Materials and Methods 2.1. Study Design and Participants, and lines 115 to 116.
Comments 6: It is necessary to describe in detail the basis for selecting the b values in the DWI sequence and how these b values affect the measurement of IVIM parameters.
Response 6: The present work is a retrospective extension of our previously published study Diagnostics 2025, 15(10), 1260; https://doi.org/10.3390/diagnostics15101260 , in which the same patient cohort and MRI acquisition protocol were employed. The IVIM-DWI sequence used the same b-value distribution (0, 30, 50, 70, 100, 200, 500, and 1000 s/mm²). Low b-values (<200 s/mm²) were included to capture perfusion-related contributions, whereas higher b-values were required to quantify true molecular diffusion and ADC.
In the Methods section 2.2. MRI Imaging, the following sentences have been added for better descriptions of the reasoning why we selected these 7 b values: “Low b-values (<200 s/mm²) were included to capture perfusion-related contributions, whereas higher b-values were required to quantify true molecular diffusion and ADC.”
These changes can be found – page number 3, 2. Materials and Methods 2.2. MRI Imaging, and lines 146 to 148
Comments 7: It is necessary to add descriptions of the specific algorithms and parameters used in the feature extraction process.
Response 7: Thank you for the suggestion. We have updated the Materials and Methods (Extraction and Selection of Radiomic Features) to direct readers to a detailed, IBSI-compliant specification of our radiomics settings. The following sentence has been added: “The full, IBSI-compliant PyRadiomics parameterization—including preprocessing, discretization (per-map bin widths), texture-matrix settings, and filters—for each IVIM map (f, D, D) is provided in Supplementary Table S1.”
These changes can be found – page number 6, 2. Materials and 2.4 Methods Extraction and Selection of Radiomic Features, and lines 188 to 191 Table S1: Supplementary Table S1. Radiomics feature-extraction parameters per IVIM map (f, D, D*).
Note: IVIM = intravoxel incoherent motion; f = perfusion fraction; D = diffusion coefficient; D* = pseudo-diffusion coefficient; ROI = region of interest; N4ITK = N4 bias field correction (ITK); GLCM = gray-level co-occurrence matrix; GLRLM = gray-level run-length matrix; GLSZM = gray-level size-zone matrix; GLDM = gray-level dependence matrix; NGTDM = neighborhood gray-tone difference matrix; LoG = Laplacian of Gaussian; ICC = These changes can be found – page number 14, Supplementary Materials: Table S1., and lines 490 to 495
Comments 8: The discussion section should provide a more detailed comparison of the results of this study with those of other related studies, especially those using traditional MRI or PET.
Response 8: Thank you for your suggestion. In the Discussion section, we have included recent studies comparable to our presented work.
“Our single-time study also produces a higher prediction performance compared that of a conventional MRI used in a follow-up study (AUC of 0.857) [61]. This highlights the potential our approach to be used for predicting the type of MS lesions in follow-up studies.”
These changes can be found – page number 12, Discussion section, and lines 387 to 390
In the Discussion section, we have included recent studies comparable to our presented work.
“A study that combined PET and MRI with automatic lesion segmentation had been shown to be powerful in predicting the disease annual relapse rate (AUC of 0.96) [62]. Our approach used a single non-ionizing MR imaging technique, which is more commonly used in neurological assessment for MS patients, and radiomics to predict EDSS. However, the utility of our approach to predict disease relapses has not been measured; this will be addressed in the future.”
These changes can be found – page number 12, Discussion section, and lines 416 to 421
Comments 9: The conclusion section should be more concise, directly answering the research questions and emphasizing the main findings of the study.
Response 9: Thank you for your comment. We agree that the conclusion is lengthy and should be more concise. “In conclusion, we showed that IVIM radiomics provided important features that can be utilised for characterising MS lesions and predicting patient disability with high accuracy. A combination of IVIM radiomics and machine learning was highly predictive of the EDSS and mobility status in patients with relapsing–remitting MS.”
These changes can be found – page number 14, conclusion section, and lines 484 to 487
Comments 10: Avoid repeating the content of the introduction or methods section and highlight the contribution and significance of the research.
Response 10: Thank you for your comment. We have removed the following contents from the Conclusion section: By addressing current limitations and pursuing the research directions outlined, future work can fully establish the role of IVIM radiomic imaging in enhancing the diagnosis, prognostication, and management of multiple sclerosis. The encouraging results presented here lay the groundwork for such advancements, highlighting the considerable promise of radiomics-driven, non-invasive lesion assessment in neurology.
We have reviewed the manuscript again to confirm no repeating contents in the Introduction or Methods section.
4. Response to Comments on the Quality of English Language |
||||||||||||||||||||||||||||||||||||||||||||||||||||||||||||||||||||||||||||||||||||||||||||||||||||||||||||
|
Point 1: Figures and tables can be improved |
||||||||||||||||||||||||||||||||||||||||||||||||||||||||||||||||||||||||||||||||||||||||||||||||||||||||||||
|
Response 1: We wish to note that the manuscript underwent MDPI’s professional proofreading service prior to initial submission. A second round of professional language editing has now been performed after making revisions according to the reviewers’ comments, ensuring improved clarity and consistency. To enhance the presentation of the manuscript, the version of each figure with high definition will be uploaded when resubmitting the paper.
|
Round 2
Reviewer 2 Report
Comments and Suggestions for Authors
thanks for your revisions.i have no further comments.